# Genome-wide identification and expression profiling of durian CYPome related to fruit ripening

**Nithiwat Suntichaikamolkul**[1], **Lalida Sangpong**[1], **Hubert Schaller**[2], **Supaart Sirikantaramas**[1,3]*

**1** Molecular Crop Research Unit, Department of Biochemistry, Faculty of Science, Chulalongkorn University, Bangkok, Thailand, **2** Institut de Biologie Moléculaire des Plantes, Centre National de la Recherche Scientifique, Université de Strasbourg, Strasbourg, France, **3** Omics Sciences and Bioinformatics Center, Chulalongkorn University, Bangkok, Thailand

* Supaart.S@chula.ac.th

**Data Availability Statement:** The nucleotide sequences reported in this article have been submitted to Genbank under accession numbers [PRJNA683229 and PRJNA732556].

## Abstract

Durian (*Durio zibethinus* L.) is a major economic crop native to Southeast Asian countries, including Thailand. Accordingly, understanding durian fruit ripening is an important factor in its market worldwide, owing to the fact that it is a climacteric fruit with a strikingly limited shelf life. However, knowledge regarding the molecular regulation of durian fruit ripening is still limited. Herein, we focused on cytochrome P450, a large enzyme family that regulates many biosynthetic pathways of plant metabolites and phytohormones. Deep mining of the durian genome and transcriptome libraries led to the identification of all P450s that are potentially involved in durian fruit ripening. Gene expression validation by RT-qPCR showed a high correlation with the transcriptome libraries at five fruit ripening stages. In addition to aril-specific and ripening-associated expression patterns, putative P450s that are potentially involved in phytohormone metabolism were selected for further study. Accordingly, the expression of *CYP72*, *CYP83*, *CYP88*, *CYP94*, *CYP707*, and *CYP714* was significantly modulated by external treatment with ripening regulators, suggesting possible crosstalk between phytohormones during the regulation of fruit ripening. Interestingly, the expression levels of *CYP88*, *CYP94*, and *CYP707*, which are possibly involved in gibberellin, jasmonic acid, and abscisic acid biosynthesis, respectively, were significantly different between fast- and slow-post-harvest ripening cultivars, strongly implying important roles of these hormones in fruit ripening. Taken together, these phytohormone-associated P450s are potentially considered additional molecular regulators controlling ripening processes, besides ethylene and auxin, and are economically important biological traits.

## Introduction

Durian (*Durio zibethinus* L., see S1 Fig), well known as the king of fruit, is an important economic fruit crop in Southeast Asia. Besides having rich custard and almond flavors, the aril flesh also contains a number of different bioactive compounds that are beneficial to human

**Funding:** This research was funded by Chulalongkorn University (grant number GRU6407023008-1) to S.S. S.S. and H.S. acknowledge financial support from Franco-Thai Mobility Programme/PHC SIAM 2021-2022 international exchange program N°46969PF. N.S. is supported by the Second Century Fund (C2F), Chulalongkorn University. The funders had no role in study design, data collection and analysis, decision to publish, or preparation of the manuscript.

**Competing interests:** The authors have declared that no competing interests exist.

health [1, 2]. In terms of global trade, Thailand has been the top durian exporter with 65.9% market share of the 2019 global production, comprising more than 1.47 billion USD for export value (https://www.tridge.com/, search in 2021). However, this climacteric fruit has a short post-harvest life, posing problems to extending the marketing period and export to distant markets [2]. Fruit organoleptic properties such as flavor, odor, and color appearance are the main factors affected by developmental and ripening processes. Traditional breeding is often used to improve yields and fruit quality. However, this is difficult because of the long cultivation, slow reproductive cycle, and quality traits complexity of durian [2]. Therefore, in-depth knowledge of the genetic and biochemical basis of such characteristics is strongly needed. In 2017, the durian genome was sequenced [3], and since then advances in 'omics' techniques provided a better understanding of the molecular mechanisms underlying fruit ripening, including those related to sulfur metabolism [1, 4] and transcription factors [5, 6]. However, biosynthetic pathways and their regulatory mechanisms governing important traits, such as aril flesh softening, the synthesis of pigment and bioactive compounds, sugar metabolism, and phytohormone crosstalk, are still unknown in durian and should be further investigated.

Cytochrome P450-dependent monoxygenases (P450s), heme-thiolate proteins, comprise a large superfamily of enzymes that respond to environmental stimuli and developmental stages. The general mechanism of this enzyme involves the addition of an oxygen atom regiospecifically and stereospecifically [7, 8]. Plant P450s have evolved into diverse families, which usually exhibit biochemically conserved functions. Families with essential functions, such as hormone metabolism or the synthesis of biopolymers, typically show a low copy number, whereas families with adaptive functions have expanded in certain taxa [9]. Most plant P450s are anchored to the cytoplasmic surface of the endoplasmic reticulum by a hydrophobic peptide present at the N-terminus, possibly forming a transmembrane segment and targeting some P450s to plastids or mitochondria [10]. Recently expanded P450 families might have new ecological functions [7], but these are more difficult to predict than the functions of conserved P450 families. Therefore, few P450s have been functionally characterized. Mining the plant cytochrome P450 complement (CYPome) in various crops such as soybean [11], rice [12], tomato [13], and grapevine [14, 15], has revealed significant information for plant quality improvement. For example, CYP78A predominantly enlarges the pericarp and septum tissues of tomato fruits [16]. Further, the fruit ripening process is controlled by various biosynthetic pathways involved in the action of P450s. Therefore, it is hypothesized that any P450 enzyme expressed during the course of fruit development and ripening has a considerable potential to act as a key catalyst for the biosynthesis of bioactive metabolites implied in that process.

In this study, we subjected the durian genome to a comprehensive search for genes encoding P450s. Information on durian P450s was then compared to that of other fully sequenced plant genomes. Taken together with transcriptome libraries, we identified all cytochrome P450s in durian (DzP450s) and determined their expression profiles throughout five development and post-harvest ripening stages of the Monthong cultivar, a slow post-harvest ripening cultivar. Large-scale expression analyses were performed for selected *DzP450* genes using reverse transcription-quantitative polymerase chain reaction (RT-qPCR). To validate ripening-associated P450s, we determined the expression levels of the candidate *DzP450s* in the aril of fruit treated with the ethylene inhibitor (1-methylcyclopropene) and ethylene releaser (ethephon). We also compared gene expression during the ripening stages to that of another popular Thai cultivar, Phuangmanee, a quick post-harvest ripening cultivar. Finally, we investigated potential cultivar-dependent P450s that preferentially expressed in four different cultivars exhibiting different ripening behaviors. As a result, we found that the remarkable expression of P450s is potentially involved in the metabolism of phytohormones during the ripening stages of durian fruit, suggesting possible crosstalk between phytohormones that regulate fruit ripening.

# Materials and methods

## Plant materials

Durian fruit of slow post-harvest ripening cultivars (Monthong and Kanyao) and quick post-harvest ripening cultivars (Chanee, and Phuangmanee) were harvested for independent biological replicates (one fruit, separate trees) from a commercial plantation located in eastern Thailand (12°40′39.2″N 102°05′35.2″E). The number of durian fruits used depends on the individual experiments listed in the next sections. Fruit samples of a similar size and weight (~3–4 kg each) were harvested at different stages, specifically preharvest stages (immature 1 and immature 2), harvest stage (mature), and postharvest stages (midripe and ripe). To ensure that samples of the different cultivars were compared at the same ripening stage, the fruit firmness of each sample was measured as previously described [5]. Briefly, the firmness values of the immature 1, immature 2, mature, midripe, and ripe stages were in the ranges of 36–44 N, 44–47 N, 47–54 N, 2–6 N and 1–2N, respectively. For immature1 stage, fruit samples were collected and peeled at 70 days (Chanee and Phuangmanee) and 85 days (Monthong) after anthesis. For immature2 stage, fruit samples were collected and peeled at 80 days (Chanee and Phuangmanee) and 95 days (Monthong) after anthesis. For the mature stage, fruit samples were collected and peeled at 90 days (for Chanee and Phuangmanee) and 105 days (for Monthong) after anthesis. For the midripe stage, fruit samples at the mature stage were kept at 30°C for 2 days (for Chanee and Phuangmanee) and 3 days (for Monthong) and then peeled. For the ripe stage, fruit samples at the mature stage were kept at 30°C for 3 days (for Chanee and Phuangmanee) and 5 days (for Monthong and Kanyao) and then peeled. All peeled samples were immediately frozen in liquid nitrogen and stored at −80°C for transcriptome and RT-qPCR analysis.

## Ethephon and 1-MCP treatments

In total, 15 independent biological replicates (one fruit, separate tree) of the Monthong cultivar (mature stage) were used, which were then randomly separated into three treatment groups, with five fruits for the control (natural ripening), five fruits for ethephon treatment, and five fruits for 1-methylcyclopropene (1-MCP) treatment, as previously described by Khaksar et al. [5]; for the control group, the fruit samples were left at 30°C. For ethephon treatment, the cut surface of the fruit stalk was brushed with aqueous 52% (w/v) ethephon (2-chloroethylphosphonic acid, Bangkok, Thailand), at approximately 1 mL/stalk, and dried at 30°C. For 1-MCP treatment, each fruit sample was placed in a 76 L sealed container and treated with 1-MCP gas for 12 h at 30°C. 1-MCP gas was generated by adding water to a 1-MCP tablet (Xianfeng, China) at a final concentration of 1% (w/w), and 38 mL of the 1-MCP solution was immediately placed in a beaker. This resulted in a final concentration of 0.5 mL L$^{-1}$ of 1-MCP in the container. The fruit samples were stored at 30°C and 85–90% relative humidity. Fruit in all groups were peeled after 5 days (ripe stage) and immediately stored at −80°C for further analysis.

## Identification and classification of durian P450 genes

All P450 sequences of *Arabidopsis thaliana*, cotton (*Gossypium raimondii*), and cacao (*Theobroma cacao*) were obtained from the Cytochrome P450 Homepage (https://drnelson.uthsc.edu/) [17]. These sequences were used as input queries to search against the durian genome database (PRJNA400310) using TBLASTN with an e-value cut-off of 1e−5. The other parameters were set at default values. All top blast hits were collected, redundancies were removed, and protein IDs and sequences were searched in the NCBI database. Possible missing

sequences were also manually mined in the protein table for *D. zibethinus* (https://www.ncbi.nlm.nih.gov/genome/?term=Durio) by searching for "P450," "CYP," "hydroxylase," and "monooxygenase" as keywords. All collected sequences were initially classified based on protein homology against well-nomenclature P450s from cotton or cacao (https://drnelson.uthsc.edu/); families share $\geq$ 40% identity and subfamilies share $\geq$ 55% identity [18].

## Conserved motif analysis

Protein sequences of durian P450s were collected in fasta format and separated into two groups, A-type and non-A-type. MEME suit [19] was used to analyze conserved motif structures in these sequences using default parameters. The amino acid frequencies of each motif were generated in the individual profiles.

## Phylogenetic analysis

All protein sequences of durian P450s were aligned using CLUSTALW [20] with default parameters. Phylogenetic trees were constructed using maximum likelihood as a statistical method in MEGAX software [21]. Tree topology was assessed by bootstrap analysis with 100 resampling replicates. The tree was visualized and colored using Figtree (http://tree.bio.ed.ac.uk/software/figtree/).

## Transcriptome analysis of Musang king and Monthong cultivars

Expression analysis was performed to gain insight into the role of the identified DzP450 in various tissues. We used publicly available RNA-sequencing data from the MaGenDB webpage [22] to determine the gene expression of *DzP450*s in four different tissues, namely leaf, root, stem, and aril, of the Musang king cultivar (the sequenced genome). For the five ripening stages, we used our RNA-sequencing data of the Monthong cultivar [Project accession number: PRJNA683229 (Mature and Ripe stages) and PRJNA732556 (Immature1, Immature 2, and Midripe stages)]. Reference-based transcriptome analysis was performed using OmicsBox program (v1.4.1.1). The raw reads were filter using FASTQ quality check package of the program using default parameters. Then, the clean reads were aligned to the reference genome of durian cv. Musang King [3] using the STAR package (v2.7.8a) [23], and the expression value at the gene level was calculated using HTSeq (v 0.9.0) [24] with default parameters. The reads per kilobase of transcript and per million mapped reads (RPKM) method was used for normalization of the expression levels [25]. The expression profile of the genes at different stages was analyzed using time course expression analysis of maSigPro (v.1.58.0) package ($P<0.05$, R-squared cutoff 0.7). Thereafter, the normalized total read counts were used to generate a heatmap using MetaboAnalyst 5.0 (https://www.metaboanalyst.ca/) [26].

## RT-qPCR analysis

Total RNA was isolated from durian fruit aril samples using PureLink Plant RNA Reagent (Waltham, MA) following the manufacturer's instructions. Genomic DNA was removed using DNase I (Waltham, MA). The quality and quantity of RNA samples were examined using agarose gel electrophoresis and an Eppendorf BioPhotometer D30 with A260/280 and A260/230 ratios from 1.8 to 2.0 and 2.0 to 2.2, respectively, following the standard guidelines described [27]. For reverse transcription, 1 µg of total RNA was used to generate cDNA using a RevertAid First Strand cDNA Synthesis Kit (Waltham, MA), following the manufacturer's recommended protocol and the standard guidelines for reverse transcription [27]. The PrimerQuest online tool (https://www.idtdna.com/PrimerQuest/Home/Index) was used to design the

primers used in this study, which are presented in S1 Table. RT-qPCR was performed in a total volume of 10 μL containing 1 μL of diluted cDNA, 5 μL of Luna Universal qPCR Master Mix (Ipswich, MA), and 200 nM of each gene-specific primer. A Bio-Rad CFX95 Real-time System was used under the following conditions: initial activation at 95˚C for 3 min, followed by 40 cycles of denaturation at 95˚C for 15 s, annealing at 60˚C for 30 s, and extension at 72˚C for 20 s. Three independent biological replicates were used for each RT-qPCR experiment. The elongation factor 1 alpha (*EF-1α*) and actin (*ACT*) genes of durian were selected as reference genes for the normalization of RT-qPCR data according to our in-house transcriptome data of durian fruit from different cultivars, which confirmed the invariant expression levels of *EF-1α* and *ACT* under different experimental conditions [6]. Relative P450 expression levels were measured using the $2^{-\Delta\Delta CT}$ method [28] according to the average Ct values of the two reference genes [29]. The normalized Ct values of each gene are presented in S3–S5 Tables.

## Statistical analysis

Gene expression analysis for ethephon and 1-MCP treatments was performed based on five biological replicates each. Gene expression analyses for developmental and ripening stages were performed with three biological replicates each. Normalized expression values (–ΔCT) were transformed to $\log(2^{-\Delta CT})$ or $\log(2^{-\Delta\Delta CT})$. The log data were tested for normality, and a T-test, Tukey HDS, and Pearson correlation test were performed using IBM SPSS Statistics version 22 ($p = 0.05$).

## Results and discussion

### Identification, classification, and conserved motifs of durian P450s

P450 genes have been identified from the durian genome [3]. As a result, a total of 355 putative durian P450 genes with complete open reading frames was identified. All durian P450s were classified and named based on the protein sequence homology [18]; families share $\geq 40\%$ identity, and subfamilies share $\geq 55\%$ identity when aligned with the officially classified P450s that are publicly provided on the Cytochrome P450 Homepage [17]. Accordingly, durian P450 genes were distributed in 10 groups consisting of 56 families, as listed in S2 Table. This number (355) is comparable to that in cocoa (336), apple (328), and grapevine (315), higher than that in *A. thaliana* (245), banana (233), watermelon (233), cucumber (229), and strawberry (209), and lower than that in soybean (715), tomato (456), cotton (449), and rice (412). We constructed phylogenetic trees of DzP450s based on the fact that plant P450s are of two types, A-type and non-A-type [9]. As shown in Fig 1, A-type (52.4%, 186/355) consisted of the CYP71 clan and the non-A type (47.6%, 169/355) consisted of the remaining families.

 According to the distribution of P450 families across plant phylogeny, there are 10 clans in plants that are named by their lowest family numbers (CYP71, CYP72, CYP85, CYP86, CYP51, CYP74, CYP97, CYP710, CYP711, and CYP727). In the durian genome, the CYP71 clan was determined to be the largest clan and comprises 52.4% of durian P450 genes. This proportion is similar to that in *Arabidopsis* (50% A type), soybean (56.08% A type), and apple (56.10% A type). In durian, A-type P450 genes were found to be more divergent than the non-A-type. This is because most A-type genes encode plant-specific enzymes that act in the metabolism of diverse secondary metabolites that enhance plant adaptation. Non-A-type genes are mainly involved in the synthesis of hormones and other compounds related to primary metabolism in plant [30]. Considering the number of angiosperm P450s (see S2 Table), we identified one CYP92, one CYP727, one CYP733, 14 CYP736s, and 21 CYP749s in the durian genome, but these have no orthologs in *A. thaliana*, a model organism that has been extensively studied for the functions of P450 genes. This suggests an evolutionary path in the

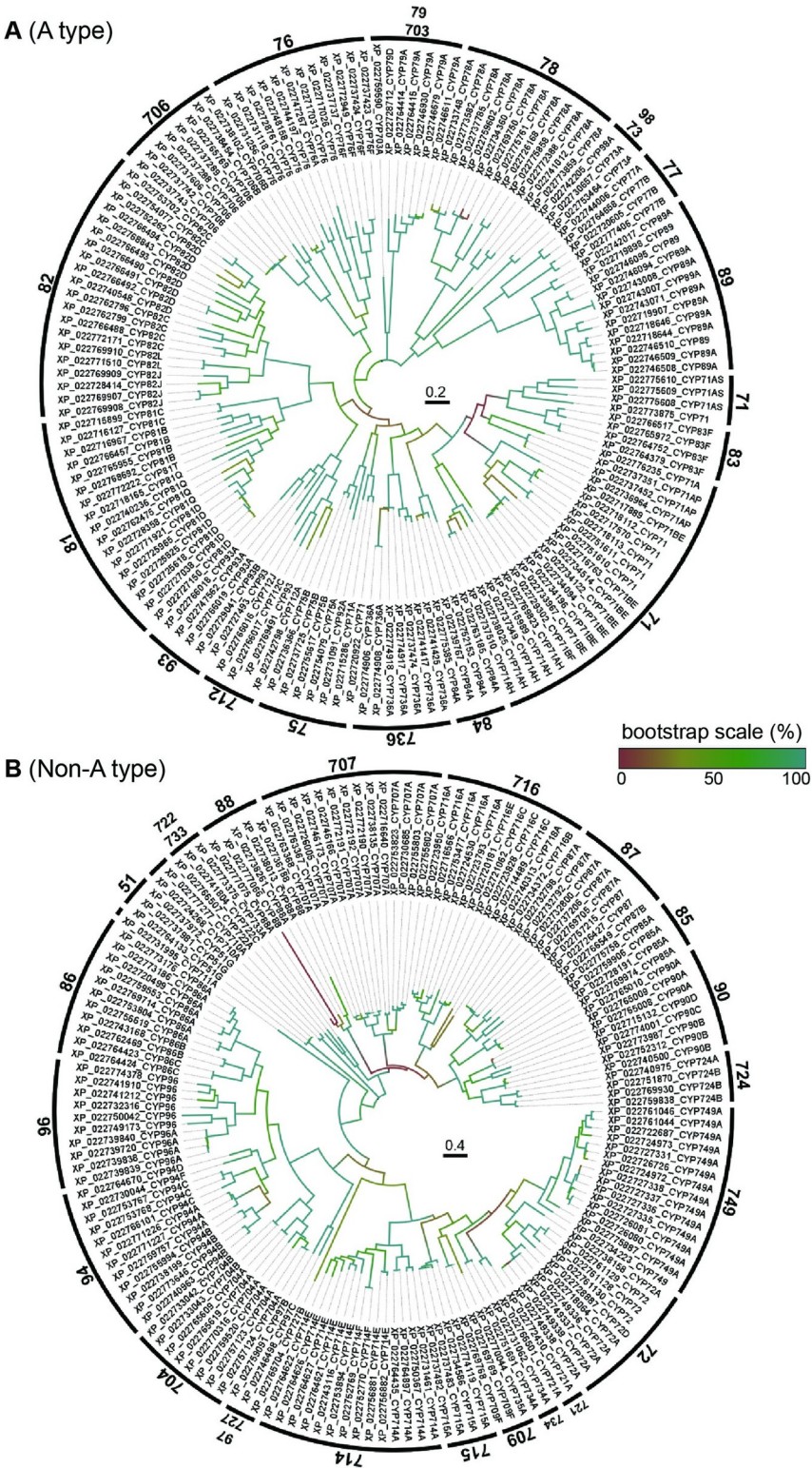

**Fig 1. Phylogenic tree of durian cytochrome P450s.** (A) A-type is represented by clan 71. (B) Non-A type is represented by the other families. The numbers on each clan indicate the family of cytochrome P450. The name of durian P450s is indicated by the protein accession number followed by the predicted (sub) family. The tree was constructed by the maximum likelihood method with 100 replicates. The colors on the trees indicate the bootstrap values (percentages) after 100 replicates. The scale bars in the circular trees represent the number of amino acid substitutions per site.

durian genome, which differs from that of the *Arabidopsis* genome. The present analysis and previously published data indicate that CYP92 exists in all higher plants and is involved in brassinosteroid biosynthesis [31]. The functions of CYP727 and CYP733 have not yet been identified. CYP736 is involved in the biosynthesis of biphenyl phytoalexins in apple [32], and CYP749 is required for herbicide tolerance in cotton [33]. To the best of our knowledge, no P450s in durian have been characterized. To further confirm this identification, a total of 355 predicted durian P450 genes was subjected to protein domain and motif analyses. As shown in Fig 2, both A- and non-A types of durian P450 proteins had structures typical of the P450 family [30], including a heme-binding region, PERF motif, K-helix region, and I-helix region. In comparison with those of Arabidopsis [30], the Glu and Arg of the K-helix, the Phe, Gly, and Cys of the heme-binding motif, and the Arg of the PERF motif were found to be respectively conserved in both plants. Meanwhile the residues of the I-helix involved in oxygen binding were slightly different.

## Gene expression analysis reveals putative *DzP450*s related to fruit ripening

The five stages of durian were divided into two main stages, developmental stages and ripening stages. Developmental stages refer to the pre-harvest to harvest stages (immature 1, immature 2, and mature) and ripening stages refer to harvest to post-harvest stages (mature, mid-ripe, and ripe). The expression levels of *DzP450*s, analyzed by RNA-seq at five stages of fruit (S1 Fig) were quantified as RPKM data. The expression signals of 44.3% (155/355) of the durian P450 genes were detected. As shown in Fig 3, the expressed P450 genes were grouped into 16 clusters, A to P, according to their expression pattern during the five stages based on ward clustering algorithm. The expression profiles of nine randomly selected *DzP450*s (from the clusters in which the expression increased during the ripening stages) were validated by RT-qPCR. These selected P450s were analyzed for their relative transcript abundance and are graphically represented in S2 Fig. Pearson correlation values (r) ranged from 0.547 to 0.983, indicating a highly positive correlation between RNA-sequencing data and RT-qPCR analysis. Therefore, our transcriptome data of the Monthong cultivar was highly accurate for examining the expression of other genes.

Considering fruit-specific P450s, gene expression of P450s across four tissues of the Musang king cultivar was determined. Accordingly, *CYP71*, *CYP72*, *CYP81*, *CYP83*, *CYP84*, *CYP88*, *CYP89*, *CYP93*, *CYP94*, *CYP96*, *CYP706*, *CYP707*, *CYP714*, *CYP718*, and *CYP749* were abundantly expressed in the aril compared to levels in the leaves, stems, and roots (S3A Fig). Taken together with the expression pattern of P450s during the five stages of the Monthong cultivar, *CYP93* was decreased, whereas *CYP81*, *CYP83*, *CYP84*, *CYP88*, *CYP89*, *CYP94*, *CYP96*, *CYP706*, *CYP707*, *CYP714*, *CYP718*, and *CYP749* increased during the five stages. Among the *CYP71* genes, the expression of some putative genes decreased, whereas some putative genes were decreased during the five stages (S3B Fig).

CYP71 is the largest P450 clan in plants, containing over half of all known cytochrome P450s identified from the plant kingdom [7, 9, 34]. Similar to a previous genomic study [3], CYP71 was upregulated in durian arils during the ripening process. Most members of this family have been functionally characterized as terpenoid oxidases, including mono-, sesqui-, and diterpenoid-modifying enzymes [35–37]. For CYP81, this family is generally involved in phenolic metabolism, the substrates of which depend on subfamilies. CYP81B and CYP81D subfamilies were upregulated during the ripening stages of the Monthong cultivar. CYP81B has been reported as involved in the in-chain hydroxylation of fatty acids in *Helianthus tuberosus* [38]. CYP81D can be induced by various exogenous stimuli, such as a number of herbicides [39–42], jasmonic acid [42–44], and salinity [45]. Therefore, CYP81B and CYP81D

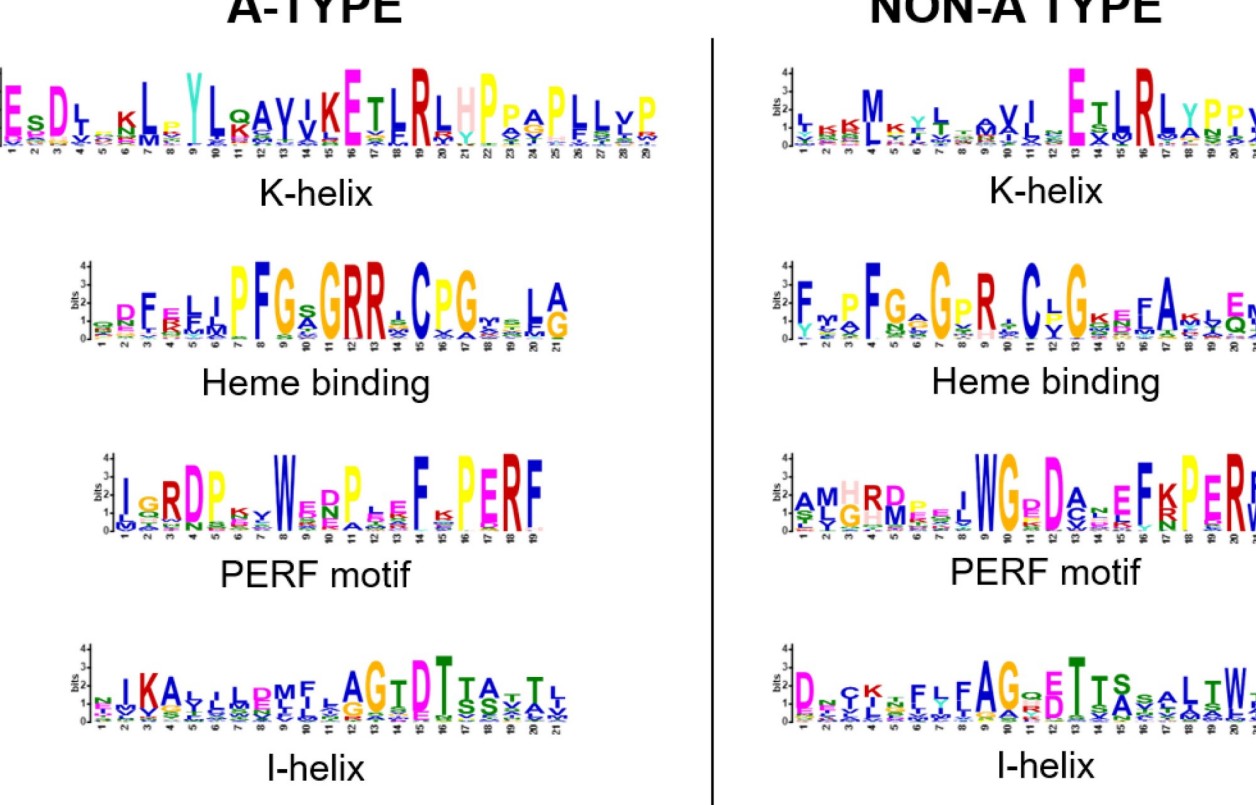

**Fig 2. Comparison of sequence logos for conserved motifs of the predicted P450 proteins, A-type and non-A type, in durian.** Sequence logos of the consensus motifs were created using MEME online software, containing stacks of letters at each position in the motif. The total height of the stack was the information content of that position in the motif in bits. The height of each letter represents the frequency of amino acids at the corresponding position.

might be involved in the biosynthesis of defense compounds against biotic and abiotic stresses during the ripening process of durian. CYP84A encodes a C-5 hydroxylase of coniferaldehyde and coniferyl alcohol, leading to syringyl lignin formation [46]. The downregulation of rice CYP84A results in altered lignin that is largely enriched in guaiacyl units, whereas the upregulation of rice CYP84A leads to enrichment in syringyl units [47], suggesting that CYP84A might be a major factor controlling lignin composition in durian fruit. CYP89A has been reported to be involved in the formation of major chlorophyll catabolites during leaf senescence in *Arabidopsis* [48], suggesting its role in pigment formation in the aril over the course of ripening. CYP93A is the ancestral group distributed in angiosperms and is involved in the biosynthesis of flavones [49]. The increased expression of CYP93A supports the occurrence of flavonoids in durian fruit [50]. CYP96A mediates omega-hydroxylation of fatty acids, which is essential for the synthesis of *Arabidopsis* cuticles [51], suggesting a role for the synthesis of aril wax during the ripening stage. CYP706 evolved from the CYP71 clan and was reported to be involved in the oxidation of both monoterpenes and sesquiterpenes [52, 53], suggesting a role for durian aril defense during the ripening stage. The biological functions of CYP718 and CYP749 remain unknown. However, CYP718 is a single-copy gene in most plant genomes (see S2 Table), indicating strong purifying selection and a possible role in durian fruit ripening. The other remaining families (CYP72, CYP83, CYP88, CYP94, CYP707, and CYP714) have been reported to be involved in phytohormone metabolism and are mainly discussed in the

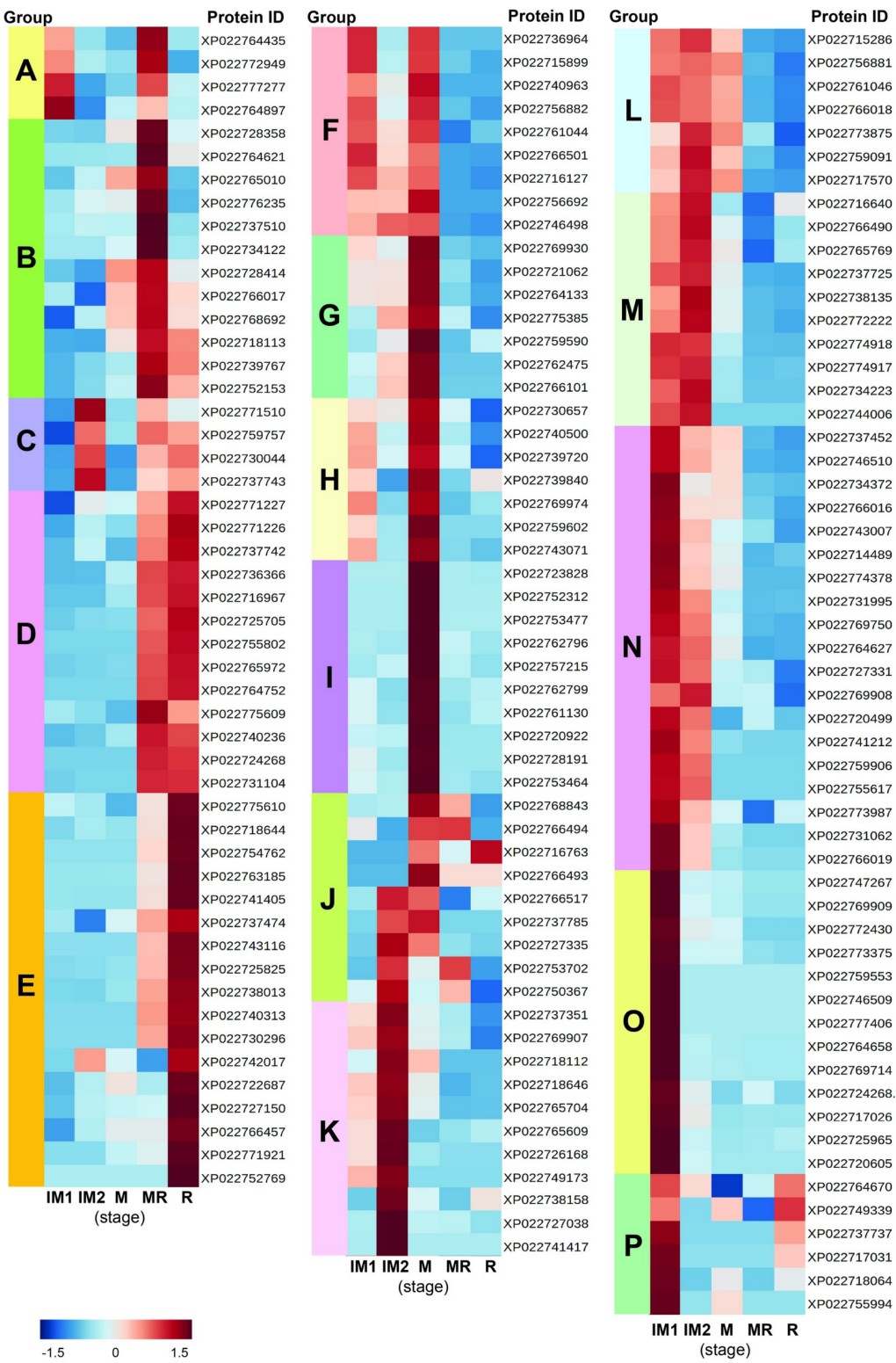

**Fig 3. RNA-seq based expression profile of durian P450s during the five stages of the Monthong cultivar.** The P450s were grouped as groups A–P based on a ward clustering algorithm using MetaboAnalyst 5.0, an open-source R-based program. Data were sum normalized, log-transformed, and auto scaled. The color scale indicates the z-score. Stage abbreviations: IM1, immature1; IM2, immature2; M, mature; MR, mid-ripe; R, ripe.

next section. It should be noted that metabolite information in durian arils is scarce and is necessary for the functional elucidation of DzCYP450s.

## Ripening-related cytochrome P450s potentially regulate phytohormones in durian fruit

**CYP72, CYP88, and CYP714 are involved in gibberellins biosynthesis.** In total, 11 CYP72s were identified in the durian genome, but only XP_022738158, which showed 82.55% identity to cotton CYP72A, was abundantly expressed in the aril of the Musang king cultivar with gradually increasing expression during the ripening stages of the Monthong cultivar (S3 Fig). CYP72A has a gibberellin (GA)-inactivating function via the 13-hydroxylation of GAs and *ent*-kaurenoic acid (KA), which abundantly occurs in *Arabidopsis* seeds, indicating that CYP72A might play a role in promoting dormancy in *Arabidopsis* [54]. Although the expression of *DzCYP72A* was not changed by 1-MCP and ethephon treatment and was not specific to either fast-ripening or slow-ripening cultivars, it was significantly higher in Phuangmanee cultivar than in the Monthong cultivar during the ripening stages (Fig 4A–4C), suggesting that it is not related to controlling ripening behavior.

Among the five *CYP88* genes found in the durian genome, XP_022738013 was highly expressed during fruit ripening. This putative gene showed 71.6% identity with cotton *CYP88A*. This subfamily catalyzes the conversion of KA to $GA_{12}$, the precursor of all GAs (Fig 5A) [55]. The expression of *DzCYP88A* was significantly suppressed in the presence of 1-MCP (Fig 4A), suggesting its significant role in the ripening process of durian fruit. Accordingly, the expression of *DzCYP88A* was significantly higher in Phuangmanee cultivar than in the Monthong cultivar during the ripening stages (Fig 4B) and in the fast-ripening cultivar (Phuangmanee and Chanee) than in the slow-ripening cultivar (Monthong and Kanyao) (Fig 4C). Taken together, DzCYP88A is potentially involved in GA biosynthesis and potentially plays an important role in accelerating the fruit ripening process in durian.

Considering the CYP714 family, five CYP714s were found in the durian genome, among which only XP022743116 was found to be expressed gradually during fruit ripening. This putative P450 showed 86.08% identity with cocoa CYP714E. The CYP714 subfamily generally has inactivating functions by oxidizing GAs on the C and D rings [55]. In contrast to *DzCYP88A* expression, *DzCYP714E* was significantly suppressed in the presence of 1-MCP and was unchanged in the presence of ethephon (Fig 4A). In addition, *DzCYP714E* significantly expressed higher in Phuangmanee than in Monthong at midripe to ripe stages (Fig 4B). However, the expression of *DzCYP714E* was not different between fast-ripening cultivars and slow-ripening cultivars (Fig 4C).

KA is a key intermediate in the biosynthesis of several terpenoids, including the phytohormone GA (Fig 5A). According to the expression profiles in durian, KA biosynthetic genes (*ent*-copalyl diphosphate synthase, *ent*-kaurene synthase, and *ent*-kaurene oxidase) were downregulated during the ripening process (Fig 5C), suggesting that an increased level of KA accumulated during the ripening process. Subsequently, an appropriate level of KA is catalyzed by DzCYP88A to produce the precursor $GA_{12}$, which is potentially oxygenated by DzCYP72A and DzCYP714A/E. As previously mentioned, these durian P450s are upregulated during the ripening process. Therefore, we propose two possible functions for oxygenated products. First, 13-hydroxylated GA is an inactivated form of GA that reduces the negative effects of bioactive GAs. This hypothesis is supported by a study showing that GAs play a role in regulating tomato fruit ripening, in which GAs act as negative regulators of ripening regulator genes, ethylene biosynthetic genes, and subsequent ripening processes [56]. Secondly, we proposed that 13-hydroxy GA might directly regulate ethylene-responsive/biosynthetic genes.

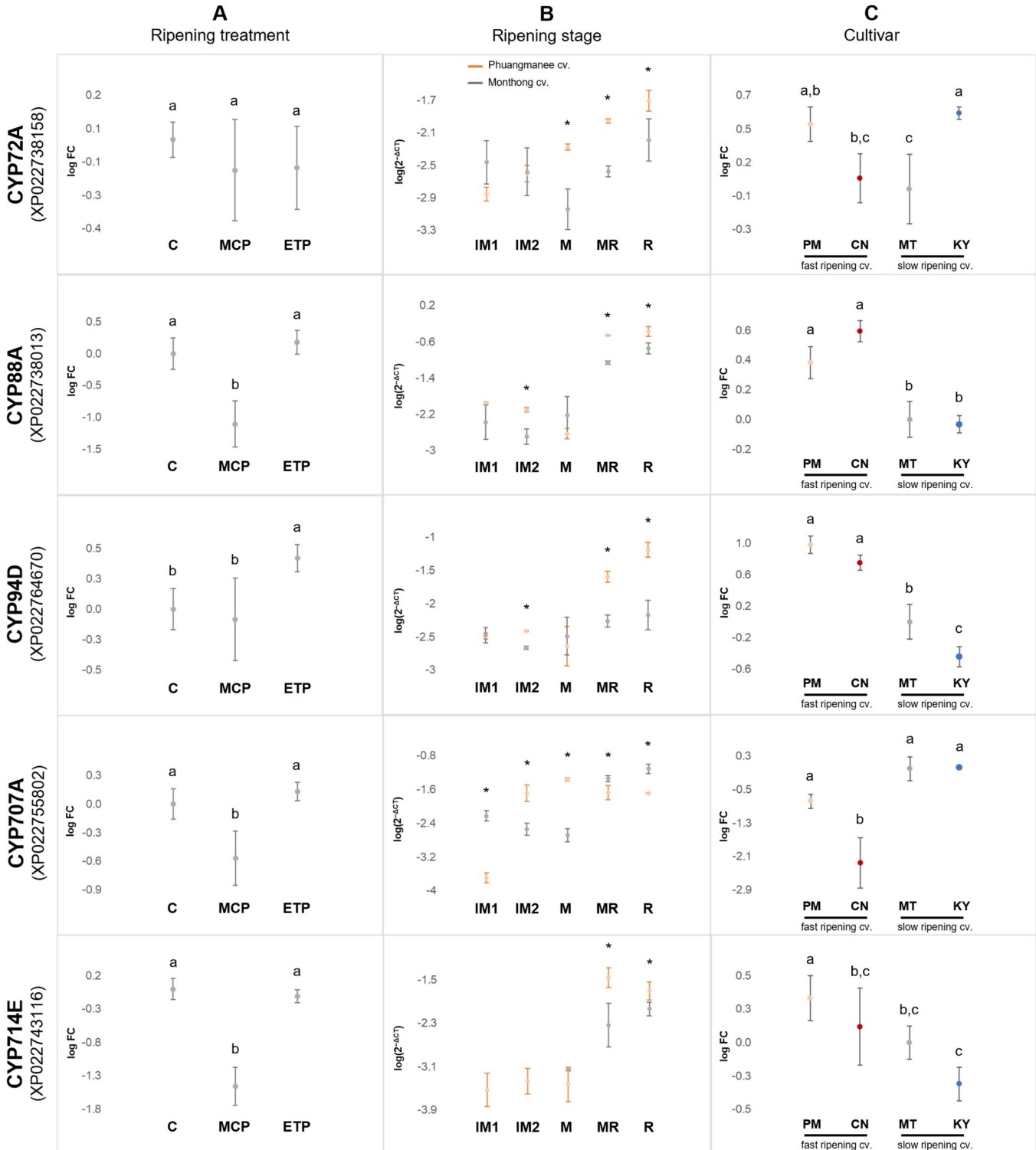

**Fig 4. Relative expression of durian P450s.** (A) Expression profiles in Monthong fruit treated with exogenous 1-methylcyclopropene (MCP) or ethephon (ETP) at the ripe stage, in relative comparison with untreated fruit (control; C). Five biological replicates were employed for each treatment. Dots and error bars represent the means and standard deviations of $\log(2^{-\Delta\Delta CT})$, respectively. Tukey's HSD tests were used for statistical calculations ($p = 0.05$). The same letter above the error bars indicates no significant difference. (B) Expression profiles during five ripening stages in Phuangmanee cultivar (orange dots) and Monthong cultivar (grey dots). Three biological replicates for each cultivar and stage were employed. Dots and error bars represent the means and standard deviations of $\log(2^{-\Delta CT})$, respectively. Asterisks indicate the significance between two cultivars at the same stage, calculated by a t-test ($p = 0.05$). The expression

of CYP714E at IM1 and IM2 stages of Monthong cultivar were undetectable by RT-qPCR. Abbreviations: IM1, immature 1; IM2, immature 2; M, mature; MR, mid-ripe; R, ripe. (C) Expression profiles in four cultivars of durian at the ripe stage (log scale). Three biological replicates of each cultivar were employed. Dots and error bars represent the means and standard deviations of log(2$^{-\Delta\Delta CT}$), respectively. Tukey's HSD tests were used for statistical calculations ($p = 0.05$). The same letter above the error bars indicates no significant difference. Cultivar abbreviations: PM, Phuangmanee; CN, Chanee; MT, Monthong; KY, Kanyao.

**CYP83 is possibly involved in auxin oxidation.** Four CYP83s were identified in the durian genome. Of these, DzCYP83F (XP_022765972), which exhibited 74.65% identity with cotton CYP83F, showed an increased expression level during fruit ripening. This family has been reported on the Cytochrome P450 Homepage [17] to be lost from several plant genomes, such as rice, tomato, apple, grape, cucumber, strawberry, and watermelon (see S2 Table), suggesting a role in plant specialized metabolism. CYP83 has been reported to regulate the level of indole-3-acetic acid (IAA) by converting indole-3-acetaldoxime to the glucosinolate pathway [57, 58]. A T-DNA insertion in the *CYP83B* gene leads to plants with a phenotype of auxin overproduction, whereas CYP83B overexpression leads to the loss of apical dominance, typical of an auxin deficit [57, 58]. In addition, in *Phaseolus lunatus*, CYP83E is involved in the biosynthesis of cyanogenic glucoside via indole aldoxime oxidation [59]. Although the biochemical function of DzCYP83F is still unknown and neither glucosinolates nor cyanogenic glucosides have been found in durian fruit, it potentially utilizes IAA as a substrate to maintain auxin homeostasis during durian fruit ripening.

**CYP94s involved in the catabolism of jasmonate and its conjugates (JAs).** Twelve CYP94s were identified in the durian genome. Among these, three DzCYP94s were specifically expressed in the aril tissue of the Musang king cultivar and were highly expressed during the ripening stages of the Monthong cultivar (S3 Fig). Two of these, XP_022759757 and XP_022771227, showed 85.27% and 84.27% identity with cocoa CYP94A. However, XP022764670 showed 84% identity with cotton CYP94D. CYP94 has been recognized to be involved in metabolic pathways of fatty acid-derived compounds, and in particular, the catabolism of bioactive JA, an oxidation derivative of polyunsaturated fatty acids. JA and its derivatives are well-known for their role in immune responses, plant growth and development, and secondary metabolism, thus allowing plants to rapidly adapt to changing environmental conditions [60]. Some reports have shown that CYP94A acts as a fatty acid hydroxylase that might be associated with cutin monomer synthesis [61–63]. Interestingly, CYP94B and CYP94C convert jasmonate-isoleucine (JA-Ile) into 12OH-JA-Ile and 12COOH-JA-Ile [64–68]. Enhanced CYP94C expression deactivates and turnovers the content of JA-Ile, conferring stress tolerance [69] and improving plant height in rice [70]. In avocado, JA-Ile was found to be increased abruptly in cold-stored fruit compared to levels in normal-stored fruit [71]. Taken together, the expression of DzCYP94B leads to the inactivation of JA-Ile, possibly resulting in the ripening process in durian. Based on the phylogenetic analysis of the plant CYP94 family, CYP94D was extended from the CYP94B/C clan (S4 Fig), which is well-recognized as a JA-Ile hydroxylase. Although CYP94D has recently been identified as a cholesterol C-26 hydroxylase in phytosteroid diosgenin biosynthesis in *Dioscorea zingiberensis* [72], the expression of DzCYP94D was induced by exogenous ethephon treatment (Fig 4A), indicating a possible role in regulating durian fruit ripening. The expression of DzCYP94D was significantly higher in the Phuangmanee cultivar than in the Monthong cultivar at mid-ripe and ripe stages (Fig 4B). In addition, it was predominantly expressed in fast-ripening cultivars (Phuangmanee and Chanee) rather than in slow-ripening cultivars (Monthong and Kanyao) (Fig 4C). It is possible that CYP94D is involved in the biosynthesis of a signal molecule that accelerates fruit ripening in durian, which remains to be further elucidated.

**CYP707 is involved in the catabolism of abscisic acid (ABA).** Among the 15 *CYP707s* found in the durian genome, only XP_022755802 was specifically expressed in the aril of the Musang king cultivar and increased throughout the ripening stages of the Monthong cultivar

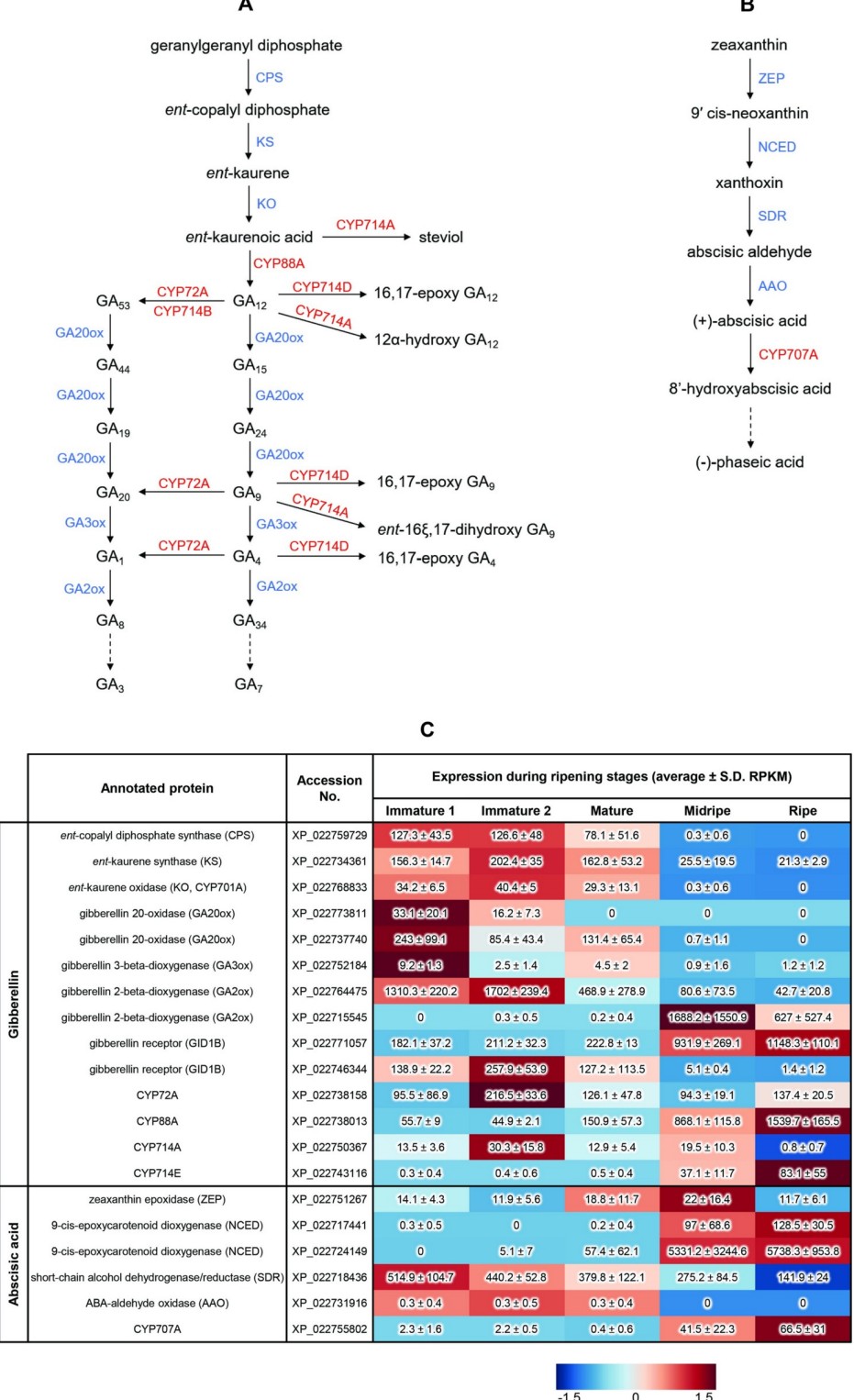

**Fig 5. Proposed reaction and RNA-seq based expression profiles involved in phytohormone biosynthesis.** (A) Gibberellin biosynthetic pathway driven by CYP72A, CYP88A, and CYP714 in durian. Enzyme abbreviations: CPS, *ent*-copalyl diphosphate synthase; KS, *ent*-kaurene synthase; KO, *ent*-kaurene oxidase; GA2ox, gibberellin 2-oxidase; GA3ox, gibberellin 3-oxidase; GA20ox, gibberellin 20-β-oxidase. Cytochrome P450s are indicated in red, whereas other enzymes are indicated in blue. (B) Proposed catalytic pathway of abscisic acid driven by CYP707A in durian.

Enzyme abbreviations: ZEP, zeaxanthin epoxidase; NCED, 9-*cis*-epoxycarotenoid dioxygenase; SDR, short-chain alcohol dehydrogenase/reductase; AAO, ABA-aldehyde oxidase. Cytochrome P450s are indicated in red, whereas other enzymes are indicated in blue. (C) Expression profiles of putative genes involved in the biosynthesis of gibberellin and abscisic acid during the ripening stages of the aril of the Monthong cultivar. The average RPKM values with standard deviations are shown in the table. The heatmap represent RPKM values, generated by MetaboAnalyst 5.0. Higher expression for each gene is presented in red; otherwise, blue was used.

(S3 Fig). This putative gene showed 90.32% identity with cocoa *CYP707A*. This subfamily has been recognized to catalyze the first committed step in the predominant ABA catabolic pathway (Fig 5B) [73] and appears to be widespread in many crop species such as rice, tomato, soybean, maize, lettuce, and wheat [74]. Constitutive expression of a *CYP707A* gene in transgenic Arabidopsis results in decreased ABA content in mature dry seeds [75]. In fact, phaseic acid (a catabolite of ABA) is unable to bind to ABA-binding proteins from apple fruit and barley aleurone layers [76, 77]. However, this molecule likely emerged in seed plants as a signaling molecule that fine-tunes plant physiology, environmental adaptation, and development [78].

Upon external 1-MCP administration to the Monthong cultivar, the expression of *DzCYP707A* was significantly suppressed (Fig 4A). In addition, the expression of this putative gene gradually increased during the ripening process of the Monthong cultivar (Fig 4B). Similar to results of previous reports, CYP707 was upregulated during the ripening process in tomato, citrus, and grape [79], as well as during fruit development in oriental melon [80]. Notably, the expression level of DzCYP707A was significantly higher in Monthong (slow-ripening cultivar) than in Phuangmanee (fast-ripening cultivar) in the late ripening stages (Fig 4B). In addition, *DzCYP707A* also expressed predominantly in Monthong and Kanyao (slow-ripening cultivars) compared to levels in Phuangmanee and Chanee (fast-ripening cultivars), implying that ABA content is probably higher in fast-ripening cultivars than in slow-ripening cultivars during the late ripening process of durian fruit (Fig 4C). Correspondingly, ABA biosynthetic genes (zeaxanthin epoxidase, 9-cis-epoxycarotenoid dioxygenase, short-chain alcohol dehydrogenase/reductase, and ABA-aldehyde oxidase) were upregulated during the ripening stages of cv. Monthong (Fig 5C). A previous report demonstrated that ABA induces the expression of ethylene-associated genes to trigger the ripening of tomato fruits [81–83]. ABA also determines fruit firmness in tomato and promotes softening, synergistically with ethylene, in banana [82, 84]. Taken together, *DzCYP707A* might play an important role in fruit ripening and accelerating the ripening process. Therefore, this gene is a potential molecular marker that requires further characterization.

## Conclusion

In climacteric fruits, including durian fruit, ethylene is the major regulator controlling fruit ripening and seems to be the most explored phytohormone. After harvest, ethylene continues to increase for a few days before declining, in parallel to the respiration of the whole durian fruit [2]. Evidence suggests that both ethylene and auxin crosstalk with each other, resulting in increased accumulation of IAA during durian ripening [5, 6]. However, the interaction between auxin and GA signaling pathways is essential for the promotion of fruit set and early fruit development stages in fleshy fruits [85]. Therefore, the active GA content might decrease during the ripening process of durian. The active GA also positively regulates the expression of ABA catabolic P450 (CYP707A) [73], which might result in decreased expression of Dz707A and elevated ABA content during the ripening stages of the fast-ripening cultivars. Accordingly, high ABA content can trigger ripening by inducing the expression of ethylene-associated genes [81–83].

Our study revealed that the durian genome has a greater number of P450 clans than various flowering plants with variable numbers of P450 genes in each clan. Phylogenetic analysis

provided information about the functional evolution of the P450 gene family in durian. The expression profile of tissue-specific P450s coupled with the expression profile of P450s during the ripening stages provides potential candidates for further study. Based on the expression profile of candidate P450s in the four durian cultivars and upon exogenous treatment with ripening regulators, we suggest that post-harvest ripening of durian fruit might be regulated by auxin, ABA, Ja-Ile, and GAs in crosstalk with ethylene. Considering the rapid climacteric nature of durian ripening, it is important to translate this knowledge to implement techniques along the supply chain to further delay or inhibit ripening. For example, an exogenous regulator of phytohormone-related P450 delays durian ripening. Not only is P450 related to fruit ripening, but this study also provides a solid foundation for the functional characterization of candidate genes with biological significance of economic importance.

## Supporting information

**S1 Fig. Morphology of durian fruit (Monthong cultivar).** (A) whole fruit. (B) peeled fruit. (C) Arils across five developmental and ripening stages. Stage abbreviations: IM1, immature 1; IM2, immature 2; M, mature; MR, mid-ripe; R, ripe.
(TIF)

**S2 Fig. Correlation between RNA-based expression profile and RT-qPCR validation of selected P450s in Monthong cultivar across five stages.** Protein ID followed by predicted P450 family are indicated above each chart. Blue bars and error bars represent the means and standard deviations of $\log(2^{-\Delta CT})$ based on RT-qPCR. Red lines and error bars represent the means and standard deviations of log-RPKM values. Three biological replicates were used for each stage. Pearson correlation (r) values are shown in green letters ($p = 0.05$). Stage abbreviations: IM1, immature stage 1; IM2, immature stage 2; M, mature stage; MR, mid-ripe stage; R, ripe stage.
(TIF)

**S3 Fig. RNA-seq based expression profile of durian P450s.** (A) Four tissues of Musangking cultivar. (B) Five ripening stages of Monthong cultivar. To simplify the heatmap, the top 70% ranked by partial least squares discriminant analysis (PLS-DA) and variable importance in projection (VIP) are shown and briefly categorized into two clusters, decreased and increased during the ripening stages. The heatmap was generated by MetaboAnalyst 5.0, an open-source R-based program. Data were sum normalized, log transformed, and auto scaled. Asterisks indicate fruit-specific P450s from the heatmap (a). The color key bars indicate the standard score (Z-score) of each gene expression level. Higher expression for each gene is presented in red; otherwise, blue was used. Stage abbreviations: IM1, immature1; IM2, immature2; M, mature; MR, mid-ripe; R, ripe.
(TIF)

**S4 Fig. Phylogeny of selected plant cytochrome P450s involved in phytohormone biosynthesis.** (A) CYP72. (B) CYP88. (C) CYPCYP94. (D) CYP707. (E) CYP714. The candidate durian P450s are highlighted in red letters. The tree was constructed by the maximum likelihood method with 100 replicates. The bars in the trees represent protein relationships of the unrooted tree. The list of protein sequences used to construct the tree is presented in S6 Table. Plant abbreviations: *At*, *Arabidopsis thaliana*; *Cl*, *Citrullus lanatus*; *Cs*, *Cucumis sativus*; *Dz*, *Durio zibethinus*; *Fa*, *Fragaria ananassa*; *Gm*, *Glycine max*; *Gr*, *Gossypium raimondii*; *Ma*, *Musa acuminata*; *Md*; *Malus domestica*; *Os*, *Oryza sativa*; *Sl*, *Solanum lycopersicum*; *Tc*, *Theobroma cacao*; *Vv*, *Vitis vinifera*.
(TIF)

**S1 Table. Primer list for RT-qPCR.**
(PDF)

**S2 Table. Comparison of P450 families in durian and selected flowering plants in the reversed order of angiosperm evolution.**
(PDF)

**S3 Table. Normalized Ct values of the genes in durian arils cv. Monthong control (natural ripening), 1-MCP treatment, and ethephon treatment.**
(PDF)

**S4 Table. Normalized Ct values of the genes in durian arils (two cultivars of each stage).**
The abbreviations N/D indicate no detectable Ct.
(PDF)

**S5 Table. Normalized Ct values of the genes in durian arils (four cultivars at ripe stage).**
(PDF)

**S6 Table. List of proteins used for construction of phylogeny of selected plant cytochrome P450s involved in phytohormone biosynthesis.**
(PDF)

## Acknowledgments

The authors are thankful to Kittiya Tantisuwanichkul, and Pinnapat Pinsorn for assistance in sample collection.

## Author Contributions

**Conceptualization:** Nithiwat Suntichaikamolkul, Supaart Sirikantaramas.

**Formal analysis:** Nithiwat Suntichaikamolkul, Lalida Sangpong.

**Funding acquisition:** Hubert Schaller, Supaart Sirikantaramas.

**Investigation:** Nithiwat Suntichaikamolkul.

**Supervision:** Hubert Schaller, Supaart Sirikantaramas.

**Validation:** Nithiwat Suntichaikamolkul, Lalida Sangpong, Hubert Schaller, Supaart Sirikantaramas.

**Visualization:** Nithiwat Suntichaikamolkul.

**Writing – original draft:** Nithiwat Suntichaikamolkul, Lalida Sangpong.

**Writing – review & editing:** Hubert Schaller, Supaart Sirikantaramas.

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
