## [Decision Letter · Decision Letter 0]

15 Sep 2021

PONE-D-21-24817

Genome-wide identification and expression profiling of durian CYPome related to fruit ripening

PLOS ONE

Dear Dr. Sirikantaramas,

Thank you for submitting your manuscript to PLOS ONE. After careful consideration, we feel that it has merit but does not fully meet PLOS ONE’s publication criteria as it currently stands. Therefore, we invite you to submit a revised version of the manuscript that addresses the points raised during the review process.

Please address all points raised by the three reviewers. In addition, please also address the following:

For expression analyses, be aware that fold-change values are not normally distributed and thus error bars should represent the upper and lower bounds of the confidence interval, rather than standard deviation/error, and statistical analyses should not assume normal distribution (or be performed on log fold change values which are normally distributed). If fold change is retained, explain clearly in the legend what comparison the fold change represents. Include (or cite another paper that includes) the primers for the reference genes and explain how both were used (e.g. was the geometric mean of both housekeeping genes used, or were these used alone as individual reference genes in different experiments?).There is displacement of some letters in gene names in S5.

We look forward to receiving your revised manuscript.

Kind regards,

Frances Sussmilch

Academic Editor

PLOS ONE

Journal Requirements:

2. In your Methods section, please provide additional location information about the commercial plantations, including geographic coordinates for the data set if available.

"This research was funded by Chulalongkorn University (grant number GRU 6203023003-1) to S.S.

S.S. and H.S. acknowledge financial support from Franco-Thai Mobility Programme/PHC SIAM 2021- 2022 international exchange program N°46969PF. N.S. is supported by the Second Century Fund (C2F), Chulalongkorn University."

Reviewers' comments:

Reviewer's Responses to Questions

**Comments to the Author**

1. Is the manuscript technically sound, and do the data support the conclusions?

Reviewer #1: Partly

Reviewer #2: Partly

Reviewer #3: Yes

2. Has the statistical analysis been performed appropriately and rigorously? 

Reviewer #1: Yes

Reviewer #2: I Don't Know

Reviewer #3: Yes

3. Have the authors made all data underlying the findings in their manuscript fully available?

Reviewer #1: Yes

Reviewer #2: No

Reviewer #3: Yes

4. Is the manuscript presented in an intelligible fashion and written in standard English?

Reviewer #1: Yes

Reviewer #2: No

Reviewer #3: Yes

5. Review Comments to the Author

Reviewer #1: In the current work, Suntichaikamolkul et al. present a deep mining of the Durian genome and transcriptome libraries, leading to the identification of P450s that are potentially involved in Durian fruit ripening. They provide insight regarding the probable activity of the different P450 clans.

Major comments:

References for the bioinformatics tools used in this analysis (SATR, tblastn, MEME, htseq clustalW etc) should be included in the materials and methods secion.

Fig 1.: The legend lacks necessary information, including whether or not the tree is rooted, which external group was used, etc. Also, the leaves could be colored to be easier to read.

Fig 2.: More detail in the legend would help to explain the figure. Is it showing the degree of conservation of the Durian P450 for each group? How did you compute the degree of conservation? Furthermore, it would be interesting to redo the same figure with Arabidopsis P450 to be able to compare the motif conservation between Durian and Arabidopsis P450s.

The heat-map in Supplementary Fig. S4 should be included in the main body of work. However, it also needs more detail—how did you define these 5 clusters and which method did you use to produce them?

In regard to Line 237 “The expression profiles of nine randomly selected DzP450s were validated by RT-qPC:” Why did you not choose at least one or two genes within the clusters that you defined in Fig. S4? The C cluste is over-represented in your figure, and we do not know if the profiles for the other cluster are consistent?

Fig. S5: The legend needs to be clearer. You included the “ratio of RPKM-normalized read counts.” Is this an RPKM or a ratio of RPKM, and, if it is a ratio, what did you use as the reference?

Sup. Fig. S6: It is difficult to understand why the RPKM is between -1 and 1 because the legend is not clear.

Line 344 "CYP83 is involved in auxin oxidation:" This part only includes published information from other species and no Durian experiments, so you cannot make a claim regarding Durian CYP83 involvement in auxin oxidation. Furthermore, the authors stated that CYP83 has been lost in several plants, so you cannot truly know what the role of this gene is in Durian, as it could have a totally new activity.

Fig. 4C: Is this a fold change or an RPKM? The column title and the color scale legend are not in agreement. Also, the blue and red labels in Fig. 4A and 4B need more explanation, along with 4C.

In the section "Durian CYPome reveals phytohormone-crosstalk in the regulation of fruit ripening," there are no results from your analysis. Therefore, this should be included in the conclusion or another part.

Line 347 “This family has been reported to be lost from several plant genomes, such as rice, tomato, apple, grape, cucumber, strawberry, and watermelon:” if it has been reported you need to include the references.

Fig. S7: You need to include the method that was used. Can you add a table with the protein ID used in these different phylogeny tree? Is the tree rooted, and, if yes, with which species? How many branches are on the tree? How did you compute these bootstraps?

Reviewer #2: In this manuscript, the authors provide valuable data regarding the molecular mechanisms putatively regulating the ripening process of durian fruit ripening that may serve as a starting point for further studies involving functional characterization of the highlighted genes presented here. A better understanding of these mechanisms could lead to different biotechnological applications, positively impacting post-harvest shelf life and other organoleptic desirable traits.

Most of my comments focus on helping to make the text clearer and more consistent. However, the authors need to pay close attention to the statements made, particularly in the results and discussion section. In general, I suggest that authors change to a more moderate writing style, especially when speculating, which is not accompanied by additional data that clearly supports the speculations.

General comments:

-When the function of a CYP is cited, be careful to always mention in what species this function was studied.

-The expression “ripening stages” used in lines 251, 361, 428 and others is very ambiguous and seems to be used as synonym of ripe developmental stage and late developmental stages in some context, while in the legends is used to refer together to the five ripening stages analyzed.

-According to PLOS Data policy, all the single measurements used for Figure 3 must be available in a table.

Introduction:

- Line 44: Please change “Fruit qualities such as flavor, odor, and color” by “Fruit organoleptic properties such as flavor, aroma, and color.”

- Lines 64-66: Please avoid the repetitive use of “therefore.”

- Line 78:

- Lines 82-83: I would suggest replacing the expression “ripening repressor” and “ripening activator” with the relation of both compounds with ethylene since these expressions sound too broad.

- Lines 86-89: I suggest rewriting this sentence since it is unclear where or when this P450 burst appears and how it is related to phytohormone crosstalk.

Materials and methods:

- Line 92: Please mention which cultivars correspond to slow- and quick-postharvest in one sentence and not in different sections throughout the text.

- Line 95: immature1

- Line 96: harvest stage (mature?)

- Line 98: Please mention briefly the firmness values used for defining each developmental stage.

-It sounds a bit confusing that three biological replicates were collected in the “Plant material” section. However, in the following section, it is mentioned that five biological replicates were employed. I suggest clarifying that these three biological replicates were stored at -80 °C for qRT-PCR analysis.

- Regarding ethephon and 1-MCP treatments: it is unclear the experimental design. Five replicates were used for each treatment (15 fruits in total?). It is also not mentioned the developmental stage of treated fruits.

Results:

- Please indicate in the legend of figure 3, if the data presented correspond to the mean, and if the error bars correspond to standard deviation or standard error. Besides, indicate the number of biological replicates employed.

- I do not understand the idea of Figure 3C. Could you please clarify what you expect to show/demonstrate with this figure, or explain why the difference of expression level at the R stage between PM/CN and MT/KY is biologically relevant for elucidating the putative role CYPs during the ripening of durian fruit? I think that the pattern of expression of each CYP during the ripening process is more relevant than the total level of expression at the ripe stage. For example, “gene A” has a very high expression level at the ripe stage, but this level remained invariable during the whole ripening process. On the other hand, “gene B” has a lower level of expression compared to “gene A” at the ripe stage; however, the expression of “gene B” has gradually and significantly increased/decreased throughout the ripening process. Could you support that “gene A” is more relevant for the ripening process than a “gene B” based just on their expression level at the ripe developmental stage?

- Also, regarding figure 3, please homogenize “a” as the lowest value and “b” or “c” as the highest. Please, present the data in the order that it is mentioned in the text.

- I have a concern regarding the statistical analysis. Tukey HSD assumes a normal distribution. Was it tested?

-Figure 3B, section CYP88A: Is there no difference between IM1, IM2, M, and MR? or between IM1, IM2, and R in the CYP714E section?

-Lines 311-312: With the provided data, it is impossible to demonstrate that DzCYP88A is involved in GA biosynthesis and plays an essential role in accelerating the fruit ripening process in durian. For this, at least a functional characterization of DzCYP88A must be performed, including substrate tolerance assays and up or downregulation assays by agroinfiltration. Moreover, is there any evidence of the role of GA in durian fruit ripening??

-Line 337: It seems that only CYP714E is upregulated.

-I am not an expert in durian, but as a climacteric fruit, the ripening of durian must be characterized by a burst of ethylene. How would you explain that CYP714E is upregulated during the ripening process but is negatively regulated by the addition of ethephon?

-Line 343: Is there any evidence to support this sentence?

-Line 374: Is there any evidence to support that JA-Ile is a negative regulator of durian ripening?

-Lines 414-46: I am confused by these sentences. First, you present ABA as a positive regulator of ripening in tomato and banana. However, in these two sentences, you say that CYP707A (involved in ABA inactivation) might play a role in accelerating the ripening process.

-Line 425: Is there metabolomic data available to support the affirmation that GA content decreases during durian ripening?

-Line 426: positively regulates.

-Lines 425-429: These assumptions contradict those stated previously in lines 304-312. First, you presented CYP88A as a key gene for GA biosynthesis, induced by ethephon and highly expressed at the ripe developmental stage (stage characterized by high levels of ethylene), speculating with a role of GA in the ripening process. Now, your crosstalk GA-ABA model is based on a decrease of GA content during the ripening.

Reviewer #3: Authors have identified 355 genes related to the cytochrome P450 gene family in Durio zibethinus and were further distributed in 10 groups consisting of 56 families. Additionally, they have also characterized the motifs and phylogenetic analysis. They have further performed the expression analysis and have validated some selected genes using the real-time PCR. The findings of current study represent an important step towards comprehending the molecular regulation of CYPs related to durian fruit ripening. However, the main problem was in the introduction and result and discussion parts. I wrote some my suggestions about this sections;

1. Introduction part; Authors should explain the significance of this study at the end of Introduction part.

2. Materials and methods part; Authors should explain Statistical Analysis in the Materials and methods part.

3. Lines 169-170, 174,…… When providing location details for a vendor, both city and country are mentioned (city, country). In case of US-based vendors, city, state suffice.

4. For the gene family expansion and evolution of novel functions, gene duplication and divergence are essential steps in the plant genome. Authors should add these researches related to Gene Duplication and Syntenic Analysis in revised manuscript.

5. Authors should carefully recheck the manuscript for scientific styles. There are some grammar mistakes; a detailed revision for English is necessary.

6. Authors should consider adding one short paragraph with a conclusion.

7. In the result and discussion section, the authors should add or discuss the following research papers:DOI：10.1007/s13258-013-0170-9 (Genome-wide identification, annotation and characterization of novel thermostable cytochrome P450 monooxygenases from the thermophilic biomass-degrading fungi Thielavia terrestris and Myceliophthora thermophila); DOI: 10.3389/fgene.2020.00044 (The Cytochrome P450 Monooxygenase Inventory of Grapevine (Vitis vinifera L.): Genome-Wide Identification, Evolutionary Characterization and Expression Analysis); DOI：10.1186/s12864-017-4425-8 (Global identification, structural analysis and expression characterization of cytochrome P450 monooxygenase superfamily in rice)

6. PLOS authors have the option to publish the peer review history of their article (what does this mean?). If published, this will include your full peer review and any attached files.

Reviewer #1: **Yes: **Axel Poulet

Reviewer #2: No

Reviewer #3: No

---

## [Author Response · Author response to Decision Letter 0]

23 Oct 2021

We have attached the response table already.

Table of response to the editor’s comments

Editor’s comments and Authors’ responses

1. For expression analyses, be aware that fold-change values are not normally distributed and thus error bars should represent the upper and lower bounds of the confidence interval, rather than standard deviation/error, and statistical analyses should not assume normal distribution (or be performed on log fold change values which are normally distributed). If fold change is retained, explain clearly in the legend what comparison the fold change represents. 

Response - Following the Editor’s comment, the expression analyses have been transformed to log scale and tested for the normal distribution (Kolmogorov-Smirnov and Shapiro-Wilk methods, p=0.05). We have re-calculated the statistical significances and found some slight changes (compared to the previous data set) then we have updated in the main text. In addition, we have been addressed statistical information in the figure legends. 

2. Include (or cite another paper that includes) the primers for the reference genes and explain how both were used (e.g. was the geometric mean of both housekeeping genes used, or were these used alone as individual reference genes in different experiments?). 

Response - Following the Editor’s comment, the reference papers for housekeeping genes and calculation method have been added in the materials and methods section (line 195) and Table S2.

3. There is displacement of some letters in gene names in S5. 

Response - Following the Editor’s comment, the figure S5 has been improved and renamed as S4 Fig.

Table of response to the reviewer’s comments

Reviewer’s comments Authors’ responses

Reviewer #1 

1. References for the bioinformatics tools used in this analysis (SATR, tblastn, MEME, htseq clustalW etc) should be included in the materials and methods secion. 

Response - Following the reviewer’s comment, we added the references for MEME (line 149), clustalW (line 153), STAR (line 168), HTseq (line 169), and RPKM normalization (line 171).

2. Fig 1.: The legend lacks necessary information, including whether or not the tree is rooted, which external group was used, etc. Also, the leaves could be colored to be easier to read. 

Response - Following the reviewer’s comment, we added the sentence “The bars in the trees represent protein relationships of the unrooted tree” in the legend of Fig 1 (line 223). All sequences from durian genome were blasted to the P450 data of a model plant (Arabidopsis thaliana) and malvaceae plants (cotton and cocoa) that share the same family with durian. According to the criteria of the P450 Nomenclature Committee (David Nelson: dnelson@uthsc.edu), we only selected durian sequences that share ≥ 40% identity and consist of major domains of the P450 family. We then unofficially named all putative durian P450s based on the Nelson’s criteria that family share ≥ 40% identity and subfamilies share ≥ 55% identity. Therefore, external P450s may not necessarily be included in these phylogenic trees because durian P450s were generated by alignment with P450s of model plants and we just need to study their distribution of each P450 family in comparison with other plants. In addition, we use the color on the branches to represent the bootstrap percentage. Adding color to the leaves might make it too colorful and make it difficult for the readers. Therefore, we would like to retain the original figure. Please consider our justification.

3. Fig 2.: More detail in the legend would help to explain the figure. Is it showing the degree of conservation of the Durian P450 for each group? How did you compute the degree of conservation? Furthermore, it would be interesting to redo the same figure with Arabidopsis P450 to be able to compare the motif conservation between Durian and Arabidopsis P450s. 

Response - Following the reviewer’s comment, we have added more detail in the legend of Fig 2 (line 249-253), indicating the program used to compute the degree of conservation. Generally, Arabidopsis is a model for plant P450 study, and its conserved motifs have been well studied (Bak et al, 2011., in Arabidopsis book). Therefore, it may be redundant to recreate conserved motifs of Arabidopsis. However, regarding the reviewer’s comment, we have mentioned the comparison of conserved motifs between durian and Arabidopsis in line 245-248.

4. The heat-map in Supplementary Fig. S4 should be included in the main body of work. However, it also needs more detail—how did you define these 5 clusters and which method did you use to produce them? 

Response - Following the reviewer’s comment, we have moved Fig. S4 to the main body as Fig 3. We have regrouped the P450s in this Fig 3 (from 5 to 16 groups) based on Ward’s clustering algorithm which was computed by MetaboAnalyst 5.0. More details have been added in the legend of Fig 3.

5. In regard to Line 237 “The expression profiles of nine randomly selected DzP450s were validated by RT-qPC:” Why did you not choose at least one or two genes within the clusters that you defined in Fig. S4? The C cluster is over-represented in your figure, and we do not know if the profiles for the other cluster are consistent? 

Response - This is because we like to focus on ripening -related P450s of durian. As a result, most of the randomly selected P450s came from groups that tended to highly express during the ripening stages. We have added the phrase "..from the clusters in which the expression increased during the ripening stages.. " to the sentence (line 263-264). 

In addition, our previous publications on durian transcription factors and other flavor-related genes demonstrated the consistent of RT-qPCR and transcriptomes of which three libraries were also used in this study (Khaksar and Sirikantaramas, 2020; Sangpong et al., 2021). We hope you accept our justification.

6. Fig. S5: The legend needs to be clearer. You included the “ratio of RPKM-normalized read counts.” Is this an RPKM or a ratio of RPKM, and, if it is a ratio, what did you use as the reference? 

Response - We have changed to log (2−ΔCT) and log(RPKM) and added information in the legend of Fig S5 (now S4 Fig).

7. Sup. Fig. S6: It is difficult to understand why the RPKM is between -1 and 1 because the legend is not clear. 

Response - The color key bars indicate normalized standard score (Z-score) of each gene expression level. The high-to-low expression level for each gene at various stages was presented in red-to-blue, respectively. We have added more details in the legend of Fig S6 (now S5 Fig).

8. Line 344 "CYP83 is involved in auxin oxidation:" This part only includes published information from other species and no Durian experiments, so you cannot make a claim regarding Durian CYP83 involvement in auxin oxidation. Furthermore, the authors stated that CYP83 has been lost in several plants, so you cannot truly know what the role of this gene is in Durian, as it could have a totally new activity. 

Response - Following the reviewer’s comment, we have changed the subtitle to be “CYP83 is possibly involved in auxin oxidation”.

9. Fig. 4C: Is this a fold change or an RPKM? The column title and the color scale legend are not in agreement. Also, the blue and red labels in Fig. 4A and 4B need more explanation, along with 4C. 

Response - The colors indicate the expression level of each gene across five stages (higher in red, lower in blue). To simplify the biosynthetic pathways of gibberellin and abscisic acid, we highlight cytochrome P450s in red letters and other enzymes in blue letters. These explanations have been added in the fig legend (now Fig 5). 

10. In the section "Durian CYPome reveals phytohormone-crosstalk in the regulation of fruit ripening," there are no results from your analysis. Therefore, this should be included in the conclusion or another part. 

Response - Following the reviewer’s comment, the section "Durian CYPome reveals phytohormone-crosstalk in the regulation of fruit ripening" has been moved to the conclusion part.

11. Line 347 “This family has been reported to be lost from several plant genomes, such as rice, tomato, apple, grape, cucumber, strawberry, and watermelon:” if it has been reported you need to include the references. 

Response - Following the reviewer’s comment, the reference has been added (line 394).

12. Fig. S7: You need to include the method that was used. Can you add a table with the protein ID used in these different phylogeny tree? Is the tree rooted, and, if yes, with which species? How many branches are on the tree? How did you compute these bootstraps? 

Response - Phylogenetic trees were constructed using maximum likelihood as a statistical method in MEGAX software with 100 of bootstrap replication as already mentioned in the method section. Therefore, these were unrooted trees show the protein relationships between plant species. Following the reviewer’s comment, we have been added the method detail in the legend and attached the supplementary table of protein ID for construction of these trees (see S8 Table). 

Reviewer #2 

1. When the function of a CYP is cited, be careful to always mention in what species this function was studied. 

Response - Following the reviewer’s comment, the manuscript has been carefully rechecked and corrected.

2. The expression “ripening stages” used in lines 251, 361, 428 and others is very ambiguous and seems to be used as synonym of ripe developmental stage and late developmental stages in some context, while in the legends is used to refer together to the five ripening stages analyzed. 

Response - Following the reviewer’s comment, the definition of developmental stage and ripening stage has been addressed at the first paragraph of gene expression analysis (line 256-259), rechecked, and corrected. We have rechecked to avoid using ambiguous words of stages.

3. According to PLOS Data policy, all the single measurements used for Figure 3 must be available in a table. 

Response - Following the reviewer’s comment, all the single measurements used for Figure 3 is available in S7 table.

4. Line 44: Please change “Fruit qualities such as flavor, odor, and color” by “Fruit organoleptic properties such as flavor, aroma, and color.” 

Response - Following the reviewer’s comment, we have been changed to be “Fruit organoleptic properties such as flavor, aroma, and color.” (line 52)

5. Lines 64-66: Please avoid the repetitive use of “therefore.” 

Response - Following the reviewer’s comment, we have deleted the repetitive therefore in line 72-74.

6. Lines 82-83: I would suggest replacing the expression “ripening repressor” and “ripening activator” with the relation of both compounds with ethylene since these expressions sound too broad. 

Response - Following the reviewer’s comment, the expression “ripening repressor” and “ripening activator” have been replaced by “ethylene inhibitor” and “ethylene releaser”, respectively (line 90).

7. Lines 86-89: I suggest rewriting this sentence since it is unclear where or when this P450 burst appears and how it is related to phytohormone crosstalk. 

Response - Following the reviewer’s comment, the revised sentence has been addressed in line 94-97.

8. Line 92: Please mention which cultivars correspond to slow- and quick-postharvest in one sentence and not in different sections throughout the text. 

Response - Following the reviewer’s comment, cultivars correspond to slow- and quick-postharvest have been addressed in one sentence (line 100 -103).

9. Line 95: immature1 

Response - Following the reviewer’s comment, the word “immature1” has been corrected (line 105).

10. Line 96: harvest stage (mature?) 

Response - Following the reviewer’s comment, the word “mature” has been addressed (line 106).

11. Line 98: Please mention briefly the firmness values used for defining each developmental stage. 

Response - Following the reviewer’s comment, the firmness values used for defining each stage have been addressed.

12. It sounds a bit confusing that three biological replicates were collected in the “Plant material” section. However, in the following section, it is mentioned that five biological replicates were employed. I suggest clarifying that these three biological replicates were stored at -80 °C for qRT-PCR analysis. 

Response - Following the reviewer’s comment, the phrase “…stored at −80 °C for transcriptome and RT-qPCR analysis” has been added in the plant material section (line 119). 

13. Regarding ethephon and 1-MCP treatments: it is unclear the experimental design. Five replicates were used for each treatment (15 fruits in total?). It is also not mentioned the developmental stage of treated fruits. 

Response - Five replicates at mature stage were used for each treatment (15 fruits in total). After 5 days, we collected the treated samples which were then ripe stage. Following the reviewer’s comment, we have been rewriting the section Ethephon and 1-MCP treatments for greater understanding.

14. Please indicate in the legend of figure 3, if the data presented correspond to the mean, and if the error bars correspond to standard deviation or standard error. Besides, indicate the number of biological replicates employed. 

Response - Following the reviewer’s comment, the definition of bars, error bars, and the number of biological replicates have been addressed in the legend of Fig 3 (now Fig 4).

15. I do not understand the idea of Figure 3C. Could you please clarify what you expect to show/demonstrate with this figure, or explain why the difference of expression level at the R stage between PM/CN and MT/KY is biologically relevant for elucidating the putative role CYPs during the ripening of durian fruit? I think that the pattern of expression of each CYP during the ripening process is more relevant than the total level of expression at the ripe stage. For example, “gene A” has a very high expression level at the ripe stage, but this level remained invariable during the whole ripening process. On the other hand, “gene B” has a lower level of expression compared to “gene A” at the ripe stage; however, the expression of “gene B” has gradually and significantly increased/decreased throughout the ripening process. Could you support that “gene A” is more relevant for the ripening process than a “gene B” based just on their expression level at the ripe developmental stage? 

Response - We found that the expression of gene A increased throughout the ripening stages in both fast ripening cultivar (Phuangmanee) and slow ripening cultivar (Monthong). It was also seen that in midripe and ripe stages, the expression of gene A was significantly higher in fast ripening cultivar (Phuangmanee) compared to slow ripening cultivar (Monthong). This leads to doubt that “Is this gene A highly expressed only for fast-ripening cultivars?” High expression of gene A, especially in fast-ripening cultivars, could lead to the discovery of a molecular key to accelerate ripening in durian fruit. Therefore, we selected another fast-ripening cultivar (Chanee) and another slow-ripening cultivar (Kanyao). We then compared the significance of gene A expression at ripe stage between fast ripening cultivars (Phuangmanee and Chanee) and slow ripening cultivars (Monthong and Kanyao) as represented in Fig 4C.

16. Also, regarding figure 3, please homogenize “a” as the lowest value and “b” or “c” as the highest. Please, present the data in the order that it is mentioned in the text. 

Response - Following the reviewer’s comment, these letters have been rearranged in Fig 3 (now Fig 4).

17. I have a concern regarding the statistical analysis. Tukey HSD assumes a normal distribution. Was it tested? 

Response - Following the reviewer’s comment, we have transformed the data to log scale and tested for normal distribution (Kolmogorov-Smirnov and Shapiro-Wilk method, p = 0.05) before the statistical analysis.

18. Figure 3B, section CYP88A: Is there no difference between IM1, IM2, M, and MR? or between IM1, IM2, and R in the CYP714E section? 

Response - For Fig 3B (now Fig 4B), we have employed t-test (p = 0.05) to calculate the significance between the two cultivars at each ripening stage. The significant differences are indicated by asterisks above each stage.

19. Lines 311-312: With the provided data, it is impossible to demonstrate that DzCYP88A is involved in GA biosynthesis and plays an essential role in accelerating the fruit ripening process in durian. For this, at least a functional characterization of DzCYP88A must be performed, including substrate tolerance assays and up or downregulation assays by agroinfiltration. Moreover, is there any evidence of the role of GA in durian fruit ripening?? 

Response - Cytochrome P450s with the same subfamily (>55% identity) generally share the same or similar functions. CYP88A encodes ent-kaurenoic acid oxidase and produces GA12 which is the key intermediate for gibberellin biosynthesis. In 1992, Mamat and Wahab detected the presence of endogenous GA4 during the development of durian fruit by GC-MS. The application of GA4 to the pedicel of durian fruits at week 6 after anthesis prevented fruit drop, enhanced fruit development, increased fruit size and weight. Although there is no evidence on how GAs are related to durian ripening, GAs have been found to decrease before ethylene production in climacteric fruit such as tomato. In addition, exogenous GAs resulted in slower fruit ripening due to reduced ethylene content and downregulated ethylene synthesis genes. However, we are functional characterizing the CYP72, CYP88, and CYP714 in model plant and planning to elucidate GA profiles of each stage of durian arils by metabolomic approach. We hope to publish this GA work in the next article. Following the reviewer’s comment, the sentence “DzCYP88A is involved in GA biosynthesis and plays an essential role in accelerating the fruit ripening process in durian” has been adjusted to “DzCYP88A is potentially involved in GA biosynthesis and potentially plays an essential role in accelerating the fruit ripening process in durian” (line 352).

20. Line 337: It seems that only CYP714E is upregulated. 

Response - CYP72, CYP88, and CYP714 are upregulated during the ripening stages (see Fig 4B). 

21. I am not an expert in durian, but as a climacteric fruit, the ripening of durian must be characterized by a burst of ethylene. How would you explain that CYP714E is upregulated during the ripening process but is negatively regulated by the addition of ethephon? Response - We would like to thank the reviewer for raising this question. We rechecked the information carefully and found an error. In fact, the expression of CYP714E was unchanged when treated with ethephon but was significantly down regulated by 1-MCP. We have edited the fig4 and the content in the main text.

22. Line 343: Is there any evidence to support this sentence? 

Response - This is our hypothesis based on the fact that the most active forms are dihydroxylated gibberellins such as GA3. Following the reviewer’s comment, we have added the word “proposed” into the sentence (line 388).

23. Line 374: Is there any evidence to support that JA-Ile is a negative regulator of durian ripening? 

Response - Although there is no evidence in durian, it has been reported that JA-Ile level was significant higher in 4 °C-stored Avocado than Avocado stored room temperature and JA-Ile biosynthetic gene is induced by auxin, suggesting a role for ripening-inhibition (Vincent et al., 2020).

Regarding the reviewer’s comment, we have adjusted the sentence to "the expression of DzCYP94B leads to the inactivation of JA-Ile, possibly resulting in the ripening process in durian" (line 420-421).

24. Lines 414-46: I am confused by these sentences. First, you present ABA as a positive regulator of ripening in tomato and banana. However, in these two sentences, you say that CYP707A (involved in ABA inactivation) might play a role in accelerating the ripening process. 

Response - ABA induces the expression of ethylene associated genes to trigger the ripening of tomato fruits. When there is sufficient ethylene level for the ripening process, then the level of ABA should be reduced by the increased expression of the CYP707A. As shown in Fig 4, Monthong has a higher expression of CYP707A (low ABA, low ethylene) at MR and R stages compared to Phuangmanee, resulting in slower ripening of Monthong.

25. Line 425: Is there metabolomic data available to support the affirmation that GA content decreases during durian ripening? 

Response - We are planning to elucidate GA profiles of each stage of durian fruit by metabolomic approach and planning to publish this GA work in the next article. Although there is no evidence in durian, studies in other climacteric fruits and associated gene expression in durian suggest that GA content is reduced during the ripening process of durian. Following the reviewer’s comment, the sentence “the active GA content may decrease during ripening process of durian” has been replaced (line 470-471).

26. Line 426: positively regulates. 

Response - Following the reviewer’s comment, the word “positively” has been added in the sentence (line 471).

27. Lines 425-429: These assumptions contradict those stated previously in lines 304-312. First, you presented CYP88A as a key gene for GA biosynthesis, induced by ethephon and highly expressed at the ripe developmental stage (stage characterized by high levels of ethylene), speculating with a role of GA in the ripening process. Now, your crosstalk GA-ABA model is based on a decrease of GA content during the ripening. 

Response - There are at least three P450s involved in the biosynthesis of gibberellins. The first P450 is CYP88, which is involved in the biosynthesis of the intermediate GA12 (well-recognized as inactive form), the key precursor producing more than a hundred of gibberellins. The other two P450s, CYP72 and CYP714, are normally involved in the conversion of active GAs to inactive GAs, resulting in a decrease in active GA level. Regarding the reviewer’s comment, the word “the active GA” has been added in the sentence (line 471).

Reviewer #3 

1. Introduction part; Authors should explain the significance of this study at the end of Introduction part. 

Response - Following the reviewer’s comment, the significance of study has been added at the end of introduction part (line 94-97).

2. Materials and methods part; Authors should explain Statistical Analysis in the Materials and methods part. 

Response - Following the reviewer’s comment, the section of statistical analysis has been added in the materials and methods part (line 197-203).

3. Lines 169-170, 174,…… When providing location details for a vendor, both city and country are mentioned (city, country). In case of US-based vendors, city, state suffice. 

Response - Following the reviewer’s comment, the location details for vendors have been edited.

4. For the gene family expansion and evolution of novel functions, gene duplication and divergence are essential steps in the plant genome. Authors should add these researches related to Gene Duplication and Syntenic Analysis in revised manuscript. 

Response - As we currently only have draft genome of durian, the exact number of chromosomes is still unknown, so the analysis of P450 duplication and syntenic may provide incomplete information. However, genome-wide analysis for P450 gene identification, transcriptome and gene expression studies during the developmental/ripening stages and the exogenous treatments of ripening agents are sufficient to predict gene function and sufficient to select the gene of interest for further functional characterization.

5. Authors should carefully recheck the manuscript for scientific styles. There are some grammar mistakes; a detailed revision for English is necessary. 

Response - The manuscript has already been edited by an English language editing service (the certificate has been attached). Moreover, following the reviewer’s comment, the manuscript has been rechecked and corrected.

6. Authors should consider adding one short paragraph with a conclusion. 

Response - Following the reviewer’s comment, the conclusion has been revised to be more succinct.

7. In the result and discussion section, the authors should add or discuss the following research papers:DOI：10.1007/s13258-013-0170-9 (Genome-wide identification, annotation and characterization of novel thermostable cytochrome P450 monooxygenases from the thermophilic biomass-degrading fungi Thielavia terrestris and Myceliophthora thermophila); DOI: 10.3389/fgene.2020.00044 (The Cytochrome P450 Monooxygenase Inventory of Grapevine (Vitis vinifera L.): Genome-Wide Identification, Evolutionary Characterization and Expression Analysis); DOI：10.1186/s12864-017-4425-8 (Global identification, structural analysis and expression characterization of cytochrome P450 monooxygenase superfamily in rice) 

Response - All three research papers are good articles and include gene duplication and syntenic section. As we have mentioned above, we do not yet know the exact number of chromosomes in durian so it may not be suitable for duplication and syntenic analysis at this stage. And the information we have in this study is sufficient to annotate gene function and enable the selection of candidate genes of interest for further functional characterization. Please note that the grape and rice papers were already cited in the introduction part. We did not cite the fungi paper because there are sufficient references for plant P450s in the result and discussion parts and the fungi paper may not be relevant to our study. Please consider our justification.

---

## [Decision Letter · Decision Letter 1]

10 Nov 2021

PONE-D-21-24817R1Genome-wide identification and expression profiling of durian CYPome related to fruit ripeningPLOS ONE

Dear Dr. Sirikantaramas,

Thank you for submitting your manuscript to PLOS ONE. After careful consideration, we feel that it has merit but does not yet fully meet PLOS ONE’s publication criteria as it currently stands. Therefore, we invite you to submit a revised version of the manuscript that addresses the points raised during the review process.

Please address the following points prior to resubmission:

Table S7a, S7b, S7c should be separate supp tables (S7, S8, S9).Fig 1 legend (L223 track changes version) – please reword to clarify what you mean by “bars in the trees”Fig 4Bar charts are not logical with these x axes (esp. as y intercept not at 0) – please change to an appropriate alternative (e.g. scatter charts, box plots, violin plots)Please make the x axis labels consistent (can also remove unnecessary repetition) – e.g. “Relative Expression (log FC)” together with an explanation of calculation in the methods would be sufficient.

We look forward to receiving your revised manuscript.

Kind regards,

Frances Sussmilch

Academic Editor

PLOS ONE

Journal Requirements:

Reviewers' comments:

Reviewer's Responses to Questions

**Comments to the Author**

1. If the authors have adequately addressed your comments raised in a previous round of review and you feel that this manuscript is now acceptable for publication, you may indicate that here to bypass the “Comments to the Author” section, enter your conflict of interest statement in the “Confidential to Editor” section, and submit your "Accept" recommendation.

Reviewer #1: All comments have been addressed

Reviewer #2: All comments have been addressed

Reviewer #3: All comments have been addressed

2. Is the manuscript technically sound, and do the data support the conclusions?

Reviewer #1: Yes

Reviewer #2: Yes

Reviewer #3: Yes

3. Has the statistical analysis been performed appropriately and rigorously? 

Reviewer #1: Yes

Reviewer #2: Yes

Reviewer #3: Yes

4. Have the authors made all data underlying the findings in their manuscript fully available?

Reviewer #1: Yes

Reviewer #2: Yes

Reviewer #3: Yes

5. Is the manuscript presented in an intelligible fashion and written in standard English?

Reviewer #1: Yes

Reviewer #2: Yes

Reviewer #3: Yes

6. Review Comments to the Author

Reviewer #1: In the current work, Suntichaikamolkul et al present a deep mining of the Durian genome and transcriptome libraries leading to the identification of all P450s that are potentially involved in Durian fruit ripening. They provide insight of the probable activity of the different P450s clan. In this current version all the comments made during the first round of review has been addressed.

Reviewer #2: (No Response)

Reviewer #3: (No Response)

7. PLOS authors have the option to publish the peer review history of their article (what does this mean?). If published, this will include your full peer review and any attached files.

Reviewer #1: **Yes: **Axel Poulet

Reviewer #2: No

Reviewer #3: No

---

## [Author Response · Author response to Decision Letter 1]

11 Nov 2021

Editor's comments

1. Table S7a, S7b, S7c should be separate supp tables (S7, S8, S9).

 Response: Following the Editor’s comment, the tables S7a-S7c have been changed to S7 Table, S8 Table, and S9 Table, respectively.

2. Fig 1 legend (L223 track changes version) – please reword to clarify what you mean by “bars in the trees”

 Response: Following the Editor’s comment, we have adjusted the sentence to " The scale bars in the circular trees represent the number of amino acid substFig 4

3. Bar charts are not logical with these x axes (esp. as y intercept not at 0) – please change to an appropriate alternative (e.g. scatter charts, box plots, violin plots)

Please make the x axis labels consistent (can also remove unnecessary repetition) – e.g. “Relative Expression (log FC)” together with an explanation of calculation in the methods would be sufficient.

itutions per site" instead (line 223).

 Response: Following the Editor’s comment, figure 4 has been changed. The labels of the y-axis and the legend of Fig 4 have been rewritten to be more concise.

---

## [Editor Report · Decision Letter 2]

15 Nov 2021

Genome-wide identification and expression profiling of durian CYPome related to fruit ripening

PONE-D-21-24817R2

Dear Dr. Sirikantaramas,

We’re pleased to inform you that your manuscript has been judged scientifically suitable for publication and will be formally accepted for publication once it meets all outstanding technical requirements.

Kind regards,

Frances Sussmilch

Academic Editor

PLOS ONE
---

## [Editor Report · Acceptance letter]

17 Nov 2021

PONE-D-21-24817R2 

Genome-wide identification and expression profiling of durian CYPome related to fruit ripening 

Dear Dr. Sirikantaramas:

I'm pleased to inform you that your manuscript has been deemed suitable for publication in PLOS ONE. Congratulations! Your manuscript is now with our production department. 

Kind regards, 

on behalf of

Dr. Frances Sussmilch 

Academic Editor

PLOS ONE